# Machine versus Human Attention in Deep Reinforcement Learning Tasks

**Sihang Guo**
UT Austin
sguo19@utexas.edu

**Ruohan Zhang**
Stanford University
zharu@stanford.edu

**Bo Liu**
UT Austin
bliu@cs.utexas.edu

**Yifeng Zhu**
UT Austin
yifeng.zhu@utexas.edu,

**Dana Ballard**
UT Austin
danab@utexas.edu,

**Mary Hayhoe**
UT Austin
hayhoe@utexas.edu,

**Peter Stone**
UT Austin, Sony AI
pstone@cs.utexas.edu,

## Abstract

Deep reinforcement learning (RL) algorithms are powerful tools for solving visuo-motor decision tasks. However, the trained models are often difficult to interpret, because they are represented as end-to-end deep neural networks. In this paper, we shed light on the inner workings of such trained models by analyzing the pixels that they attend to during task execution, and comparing them with the pixels attended to by humans executing the same tasks. To this end, we investigate the following two questions that, to the best of our knowledge, have not been previously studied. 1) How similar are the visual representations learned by RL agents and humans when performing the same task? and, 2) How do similarities and differences in these learned representations explain RL agents' performance on these tasks? Specifically, we compare the *saliency maps* of RL agents against visual attention models of human experts when learning to play Atari games. Further, we analyze how hyperparameters of the deep RL algorithm affect the learned representations and saliency maps of the trained agents. The insights provided have the potential to inform novel algorithms for closing the performance gap between human experts and RL agents.

## 1 Introduction

Researchers have devoted much effort to understanding deep neural networks (DNNs). Since DNNs are partially inspired by the biological nervous systems, researchers have compared these neural networks with the human brain and sensory systems by asking two typical types of questions. The first one considers representation: How similar are the visual representations learned by DNNs and humans when performing the same tasks? The second one concerns explainability: How do similarities and differences in the learned representations explain DNNs' performance on their tasks?

The first question motivates a seminal line of research that compares representations learned by DNNs with those learned by humans. In computer vision, a linear model that uses the feature map activations generated by a trained CNN can accurately predict neural activities in the early visual cortex, indicating that the two systems have learned similar visual representations [19, 78, 79]. In language learning, a similar connection is found between deep language models and cortical areas [31, 34].

35th Conference on Neural Information Processing Systems (NeurIPS 2021).

However, this type of comparison has just emerged in decision-making research [44, 13], which motivates us to compare DNNs trained for deep reinforcement learning (RL) tasks with human decision-making. We ask: Do deep RL agents and humans agree on what visual features are important? In other words, do agents pay *attention* to the same visual features as humans do?

The explainability question is motivated by the Explainable AI (XAI) for deep RL agents [29, 2, 55]. Deep RL has achieved many successes, but few of us fully understand these agents. These agents often learn a mapping from raw images to actions end-to-end where it is not clear why a particular decision is made. Our work addresses the explainability of these black-box models: Do RL agents make mistakes because they fail to attend to important visual features that matter for making the correct decision? An expert human's attention data could serve as a useful reference for identifying important visual features, which has been validated in object recognition tasks [50].

To answer these two questions, we situate our research in the domain of Atari games [6]. These games span a variety of dynamics, visual features, reward mechanisms, and difficulty levels for both humans and AIs. Two recent lines of research have made our work possible. On the one hand, a human attention dataset on Atari games has been collected using eye trackers [85] and convolution-deconvolution networks have been shown to accurately predict human attention distributions [83]. On the other hand, tools have been introduced to generate visual interpretations of RL agents [23] so one can visualize an RL agent's *saliency map* given an image, which is a topographically arranged map that assigns an importance weight to each image pixel and is often treated as the "attention map" of the RL agent. However, these tools can only provide qualitative results – we still need to manually inspect the saliency maps to interpret the performance. A useful comparison standard like human attention had not been considered hence systematic quantification was impossible.

In this work, we present a novel study that compares the RL agent's attention with both real and estimated human attention. We analyze how learning and hyperparameters of the RL algorithm affect the learned representations and saliency maps. We analyze failure and unseen states for RL agents and identify potential challenges to achieve human-level performance. We conclude by discussing insights gained for RL researchers in both cognitive science and AI.

## 2   Related Work

**Human vs. machine attention in vision and language tasks.**   Human experts' gaze is very efficient and accurate for solving vision tasks. The peak angular speed of the human eye during a saccade reaches up to 900 degrees per second [59]. This allows humans to move their foveae to the right place at the right time to attend to important features [17]. Therefore, human expert's gaze serves as a good standard in many vision-related tasks for evaluating machine attention, or as a learning target for training machine attention [57, 84]. This approach is widely used in computer vision, see Nguyen et al. [51] for a review. For example, visual saliency researchers train DNNs to predict human visual attention. Saliency research is a well-developed field and we direct interested readers to recent review papers [7, 10, 26, 9]. One such paper has compared visual saliency models with human visual attention [39]. In vision-related language tasks, such as image captioning and visual question answering, it was found that the saliency maps of DNN models are different from human attention [15, 72, 27]. Understanding and quantifying such differences have provided insights on the performance, especially in failure scenarios, of these vision-language models.

**Visual explanation for deep RL.**   Two classes of methods are widely used to generate visual interpretations of DNNs in the form of saliency maps: *gradient-based* and *perturbation-based*. Gradient-based methods compute saliency maps by estimating the input features' influence on the output using the gradient information [67, 68, 43, 82, 66, 71, 64, 12, 86]. These methods are for visualizing general DNNs but have been used to interpret deep RL agents [38, 76, 65, 35, 75]. Yet gradient-based saliency maps often lack physical meaning and could be difficult to interpret–they may highlight regions with no task-relevant objects or features [23, 25], thus are not used in our analyses.

Perturbation-based methods alter parts of the input image and measure how much the output is affected by the change. Hence there have been different methods of altering the input [81, 22, 14, 87, 58]. These methods have been applied to Atari deep RL agents and can generate qualitatively meaningful saliency maps [23, 33, 24, 56]. However, without human attention as a reference, it is difficult to quantitatively analyze these saliency maps, which motivates our work. There are also methods that

change the architecture of the deep RL network by augmenting it with an explicit artificial attention module, so that one can directly access its attention map [49, 80, 48]. Researchers have taken this approach and compared RL agents' attention with human attention [52]. However, these methods do not apply to general deep RL algorithms since they need to modify the original network architectures and retrain the new ones.

# 3    Method

We now discuss methods for modeling human attention, training RL agents, and extracting attention information from the trained RL agents. We then discuss how we select data for comparing human versus RL attention, and define the comparison metrics.

**Human attention data and model**    We use human expert gaze data from Atari-HEAD dataset [85]. The original game runs continually at 60Hz [6], a speed that is challenging even for professional gamers. Human eye movements were rushed and inaccurate at this speed and hence could not serve as a useful reference. In [85], however, the human data were collected in a semi-frame-by-frame mode. Without changing the action manipulation or the reward structures of the games, this design mimicked the frame-by-frame processing of RL agents for the human players. The players were allowed enough time to process the visual input, producing higher-quality data with more precise attention allocation and action execution. These advantages led to world expert-level performance in [85]–about 35x better than the human performance reported in previous deep RL literature [46, 28].

To get human saliency maps for the image states generated by RL agents, we need an accurate human attention model. RL agents make many more mistakes than expert humans and cannot reach the late stages of the games. This creates a state distribution that does not match human data, which contains mostly good states. Having a human attention model allows us to perform comparisons in states encountered by RL agents only, which is especially important when we later analyze RL agents' failure states.

Predicting human attention is a challenging task in Atari games. The human gaze is rarely on the player's avatar or contingency regions [6] so simple object detectors would not work. Human players often select and focus on a few among multiple visually identical objects, and divide their attention if multiple objects are relevant for decision making. Recently, variants of convolution-deconvolution network models have achieved the best results on predicting human attention [42, 83, 53, 16]. Here we followed their approach and trained gaze models from ground truth human gaze data. Given an image, the models produce a heatmap of the same dimension that predicts human attention distribution. These models have been evaluated and described in detail in previous studies (e.g., [83]), so we leave out the details in Appendix 1. As a control analysis, we include two attention models in addition to the human gaze. The first one captures the motion information, measured by Farneback optical flow between two consecutive images [20]. The second model captures salient low-level image features, including color, orientation, and intensity (weighted equally), computed by the classic Itti-Koch saliency model [32]. Implementation details of these models can be found in Appendix 1.

**Reinforcement learning agent and attention model**    As a case study, we use a popular deep RL algorithm named Proximal Policy Optimization (PPO) [63] with default hyperparameters [30]. For each experimental condition (discussed later), we train 5 models with different random seeds to capture the variance in training and ensure reproducibility. For the Atari gaming environment, we use the basic version that has no frame skipping and no stochasticity in action execution (NoFrameskip-v4 version). In order to capture variance in training and ensure reproducibility, we select six popular Atari games instead of using all Atari games and train 540 agents in total, with the same hyperpareameters except for random seeds and discount factors (see Section 4; 300 GPU days on GeForce GTX 1080/1080 Ti).

We use a perturbation-based method [23] to extract attention maps from PPO agents, which has been validated by several subsequent studies [25, 65]. The algorithm takes an input image $I$ and applies a Gaussian filter to a pixel location $(i, j)$ to blur the image. This manipulation adds spatial uncertainty to the region around and produces a perturbed image $\Phi(I, i, j)$. Denote the learned policy as $\pi$ and the inputs to the final softmax activation as $\pi_u(I)$ for image $I$ (i.e. the last latent representation). A saliency score for this pixel $(i, j)$ can be defined as how much the blurred image changes the latent

representation $\pi_u(I)$ in Euclidean space:

$$S_\pi(i,j) = \frac{1}{2} \|\pi_u(I) - \pi_u(\Phi(I,i,j))\|^2 .$$  (1)

Intuitively, $S_\pi(i,j)$ describes *how much removing information from the region around location $(i,j)$ changes the policy* [23]. In other words, a large $S_\pi(i,j)$ indicates that the information around pixel $(i,j)$ is important for the learning agent's decision making. Instead of calculating the score for every pixel, [23] found that computing a saliency score for pixel $i$ mod 5 and $j$ mod 5 produced good saliency maps at lower computational cost for Atari games. The final saliency map $P$ is normalized as $P(i,j) = \frac{S_\pi(i,j)}{\sum_{i,j} S_\pi(i,j)}$.

**Comparison metrics**   Next, we compile a set of game images to compute human and RL saliency maps. For each game, we let a trained PPO agent (with default hyperparameters) play the game until terminated, and uniformly sampled 100 images from the recorded trajectory. We will refer to this set of images as the standard image set.

We then define two metrics for comparing saliency maps when using RL data: Pearson's Correlation Coefficient (CC) and Kullback-Leibler Divergence (KL). Let $Q$ denote the human saliency map predicted by the human attention network. CC is between $-1$ and 1 captures the linear relation between two distributions $Q$ and $P$:

$$CC(P,Q) = \frac{\sigma(P,Q)}{\sigma_P \times \sigma_Q}$$  (2)

where $\sigma(P,Q)$ denotes the covariance, and $\sigma_P$ and $\sigma_Q$ are the standard deviations of $P$ and $Q$ respectively. CC penalizes false positives and false negatives equally.

However, we may not want to penalize the RL agent if it attends to regions that the human gaze model is not currently focused on. The human gaze data only reveals the "overt" attention, and humans can still pay "covert" attention to entities in the working memory [54, 62]. In other words, being attended to by the human gaze is a sufficient (but not necessary) condition for the features to be important. Thus we need a second metric that penalizes the agent only if it *fails* to pay attention to human attended regions, or equivalently, a metric that is sensitive to false negatives if we treat human attention as the ground truth. KL is an ideal candidate in this case [9]:

$$KL(P,Q) = \sum_i \sum_j Q(i,j) \log \left( \epsilon + \frac{Q(i,j)}{\epsilon + P(i,j)} \right)$$  (3)

where $\epsilon$ is a small regularization constant (chosen to be 2.2204e-16 [9]) and determines how much zero-valued predictions are penalized.

## 4   Results

To make meaningful comparisons, we first ensure that the human attention model is accurate, and that PPO agents' attention maps are consistent over repeated runs. We then compare human attention with PPO attention obtained from different learning stages and from agents that are trained with different discount factors. We then analyze PPO agents' attention in failure and unseen states. Finally, we show comparison results for other deep RL algorithms.

**Accuracy of human attention model**   We implement the convolution-deconvolution gaze prediction model [85] to generate a human saliency map for image $I_t$ at timestep $t$, given a stack of four consecutive images $I_{t-3}, I_{t-2}, I_{t-1}, I_t$ as input. We use 80% gaze data for training and 20% for testing. The model can accurately predict the human gaze. On testing data, we obtained Area under ROC Curve (AUC) score of $0.968 \pm 0.005$, CC of $0.562 \pm 0.030$, and KL of $1.411 \pm 0.114$ ($n = 6$), averaged over all games. Prediction accuracy for individual games can be found in Appendix 1 of the supplementary materials. A visualization of the gaze data and prediction results can be found in the attached video file. This accuracy is considered high in saliency research [9].

**Consistency of RL attention** We then show that saliency maps of the RL agents trained under the same experiment setting, provided by [30], are highly consistent despite the stochasticity in the training process. The stochasticity is controlled by a random seed that is used to initialize both the game environment and the network. For each game, we use 5 random seeds (0-4), train an agent using each seed, and generate 5 saliency maps with these trained agents. For each image in the standard set, we compute pair-wise CCs between the 5 saliency maps (10 CCs in total). The average value for these 10 CCs, across 100 images and 6 games, is $0.924(\pm0.001, n = 6000)$. Given such high consistency, the saliency maps we use later are the averaged results of these 5 saliency maps.

## 4.1 RL versus human attention: The effects of learning

So far we have verified that the gaze model can accurately predict human attention and that RL attention is consistent across repeated runs. We now address the primary research question: How similar are the visual features learned by RL agents and humans when performing the same task? We first study how the attention of PPO agents evolves over the training steps, compared with human attention. For each game, we saved neural network weights at different time steps during training. Then we use these saved models to generate saliency maps on the standard image set.

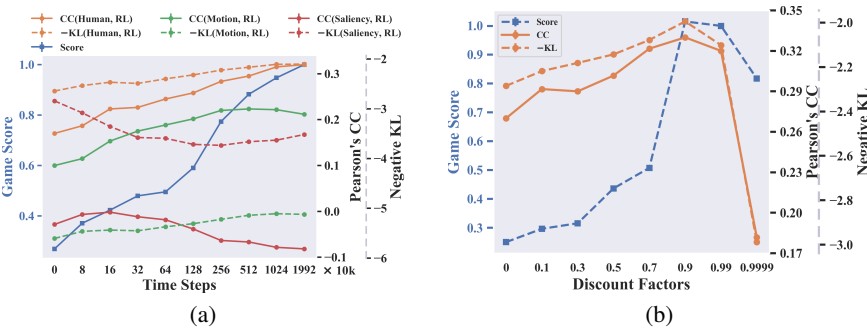

Figure 1: (a) Changes in human and RL attention similarity across learning time steps. PPO agents gradually learn to pay attention to important visual features and become more human-like. We also show comparisons between PPO agents and two control attention models: motion (optical flow) and saliency (low-level image features). The x-axis is log scale and KL values are negated for better visualization. The CC, KL, and score results for individual games, as well as more examples, can be found in Appendix 2. (b) Changes in human and RL attention similarity across different discount factors. Choosing different discount factors affects the RL agents attention and performance. The CC, KL, and score results for individual games, as well as more examples, can be found in Appendix 3.

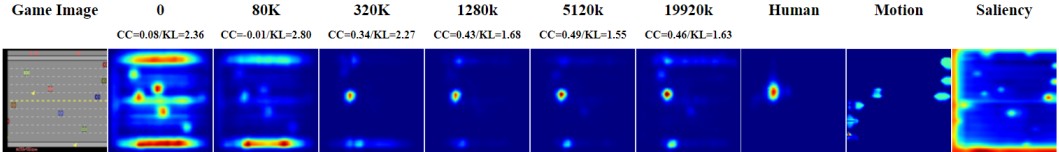

(a) Freeway: The RL agent gradually learns to focus its attention on the yellow chicken crossing the highway.

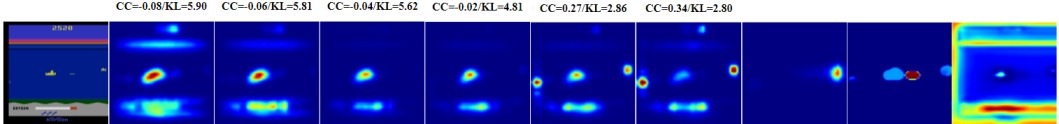

(b) Seaquest: The agent learns to attend to the yellow enemy on the right.

Figure 2: Attention of RL agents changes during learning and becomes more human-like. CC/KL values are calculated between the RL agent's attention map and the human attention map.

Fig. 1a shows the aggregated results for all games. CC values and (negative) KL values between human and RL increase during learning, indicating that the RL agents' attention gradually becomes more human-like. We visualize the change of RL attention and human attention in Fig. 2. Networks without any training (time step 0) have saliency maps that are positively correlated with humans

(CC= 0.170). For example, the second column in Fig. 2 shows that these networks are already sensitive to low-level salient visual features without any learning, which is consistent with previous findings [79, 23]. Overall, human and RL saliency maps are more positively correlated after training (CC= 0.320; CC: $r(8) = 0.992, p < 0.01$; KL: $r(8) = 0.987, p < 0.01$), meanwhile RL attention is more similar to human attention than to the control models (higher average CC/-KL values after training across all games).

We also show aggregated game performance in normalized game scores in Fig. 1a. For each game, we normalize the game scores obtained during learning (averaged over 50 episodes) by dividing them by the final scores. We find a strong positive correlation between the human CC/KL values and the game score (average Pearson's correlation coefficient of $0.813/0.790$). The correlations for all games are statistically significant ($p < 0.05$, Appendix 2), indicating that small changes in similarity with human attention are reliable predictors of performance change. For comparison, the correlation values between CC/KL and game score for motion baseline are $0.404/0.251$, for saliency baseline are $-0.258/-0.688$.

This result sheds light on an important research topic: bottom-up versus top-down attention. The two sides debate how much human or machine attention is driven by bottom-up image features captured by the saliency model [32], and how much it is driven by top-down task signals such as reward [60, 8]. Our result suggests that learning is a key factor in this debate. In the early stages, attention is more driven by image features, indicated by the higher similarity between RL attention and saliency baseline. Then top-down reward signals shape the attention during learning by making reward-associated objects more salient and irrelevant objects less salient, as shown in Fig. 2.

## 4.2 RL versus human attention: Discount factors

We then analyze how hyperparameters of the PPO algorithm affect the attention of the trained agents. We have seen that reward shapes attention during training, therefore a reasonable hypothesis is that varying reward-related parameters will likely affect attention. One of the parameters we can vary in these games is the discount factor $\gamma \in [0, 1)$, which determines how much the RL agent weighs future reward over immediate reward. The default $\gamma$ is $0.99$ for all games [30]. We train PPO agents with $\gamma \in \{0.1, 0.3, 0.5, 0.7, 0.9, 0.9999\}$ and generate saliency maps on the standard image set. For each $\gamma$ value, we normalize the game score by dividing it by the score obtained by the $\gamma = 0.99$ agent.

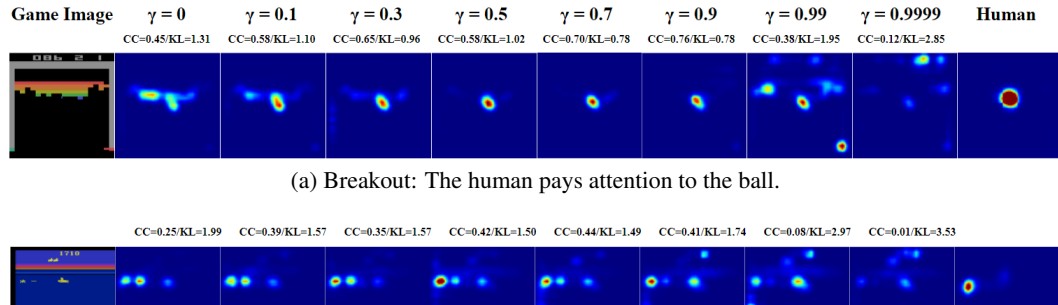

(a) Breakout: The human pays attention to the ball.

(b) Seaquest: The human pays attention to an approaching enemy from the left.

Figure 3: Effect of different discount factors on Ms.Pac-Man and Seaquest agents' attention. Agents trained with intermediate discount factor values have saliency maps that are more human-like.

The results are shown in Fig. 1b. Overall, RL attention is most similar to human's when $\gamma = 0.9$. Beyond that, RL agents learn to attend to objects that matter only in the long run. For example, Fig. 3a shows that in Breakout, RL attention is primarily on the ball (immediate effector) when $\gamma = 0.9$. When $\gamma = 0.9999$, however, it focuses on the number of lives, which does not indicate the end of the game until much later. With an intermediate value of $\gamma = 0.99$, the agent attends to both groups of objects. On the contrary, humans rarely attend to long-term objects. This is because human attention is limited, so we have to keep those objects in the working memory and revisit them at lower frequencies (e.g., [73, 37]). Therefore, these long-term objects do not show up in our human attention model, and thus the less similar attention at larger discount factors.

Although in most games, the best game performance is achieved with default $\gamma = 0.99$, Fig. 1b shows that the best performance across games is obtained at $\gamma = 0.9$. In Seaquest, the agent with $\gamma = 0.9$ achieves $15\%$ higher score than the default agent. Fig. 3b shows that with a lower $\gamma = 0.9$, the agent can focus, like humans, on an immediate threat from the left. Setting $\gamma = 0.99$ or $0.9999$ distracts the agent to attend to the oxygen bar at the bottom that is important in the long run but less urgent now. A similar result was found for Ms.Pac-Man with an $18\%$ higher score from the $\gamma = 0.9$ agent than the $\gamma = 0.99$ agent.

This result suggests that deviating from the default $\gamma = 0.99$ can lead to better performance. Atari games are episodic tasks with true $\gamma = 1$, but lower $\gamma$ values often lead to better performance in practice [63, 46][1]. Fig. 1b shows that $\gamma = 0.9999$ agents perform poorly. We have verified that all $\gamma = 0.9999$ agents (5 random seeds) converged after 200M samples, although to sub-optimal policies (Appendix 3). One reason for this deviation may be that being a little myopic helps the agent focus on the most urgent targets. This result provides another reason, from a perception perspective, for why RL agents need to adjust their planning horizon by reducing the discount factor–confirming the theoretical results provided by [36].

### 4.3    RL versus human attention: Failure states

We now turn to the second research question that concerns explainability in deep RL: Why do deep RL agents make mistakes? During training, RL agents must learn to both identify relevant objects (perception) and make the correct decisions based on that information (policy). This leads to two types of states where RL agents could make mistakes: (1) they fail to attend to the right objects, and (2) they attend to the right objects but make the wrong decisions. Since human attention can be used to identify important visual features, can we use human attention to distinguish these two types of mistakes? To answer this question, we compile a failure image set by recording the game frames right before the RL agents lose a "life" which incurs a large penalty. We locate 100 such instances for each game. Freeway is excluded since the PPO agent learned a policy that is nearly optimal.

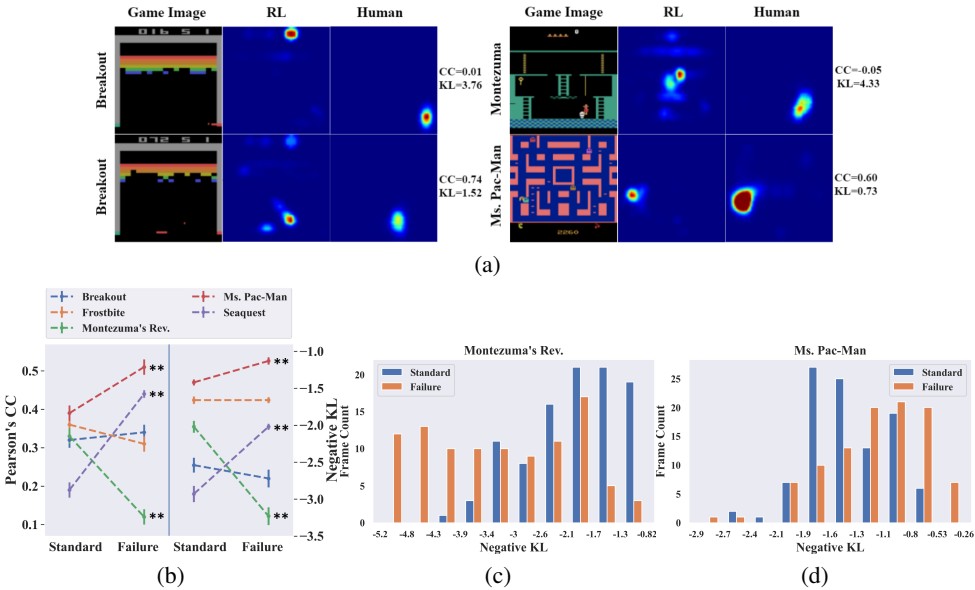

Figure 4: RL vs. human attention in states where RL agents made mistakes. (a) We show examples of RL and human saliency maps for Breakout (top: wrong attention; bottom: right attention but wrong decision), Montezuma's Revenge (wrong attention), and Ms.Pac-Man (right attention but wrong decision). (b) How attention similarities change in failure states compared to normal states. ** indicates $p \leq 0.01$. Error bars are the standard errors of the mean ($n = 100$). (c, d) Histograms of negative KL values for 100 standard vs. 100 failure states indicate that RL agents are more likely to neglect (Montezuma's Revenge) or attend to (Ms.Pac-Man) human attended regions in failure states.

---

[1]Strictly speaking, $\gamma$ is a property of the MDP, not of the agent. Performance across MDPs with different $\gamma$s are not directly comparable in this sense. Varying discount factor and clipping reward are common reward engineering designs that alter the true MDP return to achieve better performance in practice.

Fig. 4a illustrates the two types of failure states for Breakout (left column). In the first case, RL attention is extremely different from human's because it does not attend to the ball as the human model does. This lack of attention on task-relevant objects likely contributes to its failure. In the second case, the agent and the human model have highly similar attention maps because both attend to the ball. However, the agent fails to save the ball (which can still be saved), indicating that it has learned a suboptimal policy despite good perception. Following this method, we can use human attention as a reference to quantitatively interpret RL failure cases: if RL attention is in general less similar to human attention in failure than in normal states, then the model less frequently attends to task-relevant objects ("bad perception"); if it is more similar, then the model is not making the correct decisions despite the correct visual information ("bad policy").

Fig. 4b shows the quantitative results for all games. For Montezuma's Revenge, the RL attention in failure states becomes less similar to human attention compared to the standard image set, indicated by significantly decreased CC/KL values. Fig. 4c shows the (negative) KL histogram of 100 failure states vs. normal states. We can see that in the failure states the RL agents are more likely to neglect human attended regions. Fig. 4a shows that the agent fails to attend to the enemy as the human does. With the same amount of training, the Montezuma agents do not learn to identify task-relevant objects in the game. In other words, they do not learn to parse out the semantics and structures of the game. This suggests that Montezuma's Revenge is perceptually more difficult for the agents.

On the contrary, in the failure states for Ms. Pac-Man and Seaquest (Fig. 4b), attention maps of RL agents and humans are more similar than in the normal states. Fig. 4d shows the (negative) KL histogram of 100 failure states vs. normal states from Ms. Pac-Man, in which humans and RL agents are more likely to agree on the objects to attend to in failure states. An example frame is shown on the bottom of Fig. 4a: the agent attends to the Pac-Man and the enemy ghost similarly to humans, but it makes the wrong decision and runs into the ghost. Compared with Montezuma's revenge, Ms. Pac-Man is perceptually easier for the agents.

We conclude that similarity measurements with human attention may help identify and interpret the failures made by RL agents, as well as identifying tasks that are more perceptually difficult to learn for the agents.

## 4.4   RL versus human attention: Unseen data

Next, we study whether the RL agent's attention generalizes to unseen states. The unseen states are late-game states obtained from human experts' data [85] which RL agents have not encountered (i.e., above the agents' best score; score threshold for each game can be found in Appendix 5). We refer to this set as the unseen image set (100 images per game). Again, Freeway is excluded since the agent is nearly optimal.

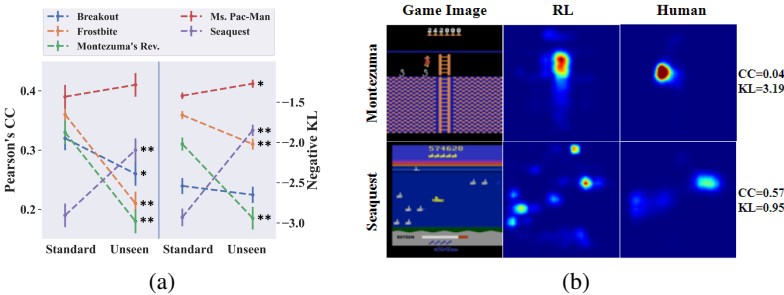

(a)          (b)

Figure 5: Human versus RL attention in states that RL agents have not seen. (a) How attention similarities change in unseen states compared to seen states in different games. (b) Unseen states for Montezuma's Revenge (top), the RL agent's attention is very different from human attention due to a new object on the left. For Seaquest (bottom), they are similar.

Fig. 5a shows the results. For Frostbite and Montezuma's Revenge, the similarities drop significantly. One reason for this drop may be the presence of new objects that the agents have never encountered in the unseen states. Fig. 5b shows an example for Montezuma's Revenge. The agents attended to the ladder, a previously seen object, but failed to attend to a new enemy object on the left like the human did. For Seaquest and Ms. Pac-Man, on the other hand, RL attention and human attention are

more similar on unseen data. Compared with the game structure in Montezuma's Revenge, Seaquest and Ms. Pac-Man do not have new objects in the unseen states–the objects only move faster and appear in larger numbers. However, this alone only supports a lack of decrease in attention similarity. We suspect the *increase* in similarity may be due to dangerous states similar to those in Section 4.3, but more controlled experiments (e.g., suggested in [3]) are required to further interpret this result. Breakout is an interesting case. The CC value drops significantly in unseen states whereas the negative KL value does not change much. The similar KL values may be attributed to the maintained attention on the ball and paddle in unseen states, whereas the decreasing CC values may be due to unseen spatial layouts of the bricks. Again, more controlled studies are required to confirm this. More examples are in Appendix 5.

The results suggest that one of the first obstacles to achieving expert human-level performance for certain games is the perception challenge–the agents need to learn to recognize, attend to, and then learn to act upon new objects. This is easy for humans due to their prior knowledge but challenging for RL agents [40, 74, 18]. For the other games without novel objects, the agent's attention is fairly generalizable and it needs to learn a good policy for challenging states.

### 4.5 Other Atari agents

The above analyses were done for a particular RL algorithm–PPO. Next we apply our method to other RL algorithms, including C51 [5], Rainbow [28], DQN [46], and A2C [45], as well as two evolutionary algorithms, GA [69] and ES [61]. We use trained models from Dopamine [11] and Atari Model Zoo [70] in which each algorithm has 3-5 trained models. There is a strong positive correlation between model performance (in terms of the game score, averaged over 50 episodes each) and similarity measurement (in terms of CC with human attention on the standard image set). The average correlation coefficient for five games is $r = 0.631$ (excluding Montezuma since most algorithms have zero scores). The results for individual games can be found in Appendix 6. This result suggests that the overall positive correlation between model performance and similarity with human attention generalizes to other deep RL algorithms. Although not performed in this work due to resource limitations, future studies should run experiments on the effect of learning and discount factors to further confirm our findings.

## 5 Discussion

We provide visual explanations for deep RL agents using human experts' attention as a reference. We have discussed how RL attention develops and becomes more human-like during training, and how varying the discount factor affects learned attention. We show that human attention can be useful in diagnosing an RL agent's failures. We also identify challenges in closing the performance gap between human experts and RL agents in the unseen states experiment. Our analysis is restricted to saliency map comparisons, but other approaches are possible for measuring the similarity of representations learned by different RL agents [77]. Our human attention models, all compiled datasets, and tools for comparing RL attention with human attention are made available for future research in this direction.

When interpreting the results, it is important to keep in mind the limitations of saliency-based explainable RL methods [71, 1]. Recent studies have shown that saliency maps are more exploratory rather than explanatory [3]. To infer causal explanations, one can start with using saliency maps to build falsifiable hypotheses, and perform counterfactual evaluations of these hypotheses, e.g., by manipulating states to generate counterfactual semantic conditions [3]. Our framework helps direct researchers' efforts towards data that deviate from human attention patterns, from which useful hypotheses are more likely to arise.

For researchers who are interested in RL algorithms, we have gained at least three important insights. First, since the task performance and similarity to human attention are highly correlated, one could use human attention as prior knowledge to guide the learning process of RL agents, e.g., by encouraging them to attend to the correct objects early in the learning process. This could be especially helpful for games like Seaquest, in which the agent has not learned to focus on the right object after 5120k time steps (Fig. 2). Two human studies using Atari games suggested that prior knowledge, such as perceptual prior, is why humans learn faster and better in these games [74, 18]. Then the next question is how to incorporate human attention into DNNs–which has been studied in several computer vision

tasks [57]. Our results indicate that humans typically attend to fewer regions than RL agents do. Therefore a desirable loss function should encourage the agents to focus on the regions that humans attend to, but would not penalize the agents for attending to more regions [62].

Second, our results provide visual explanations for the agents' performance when varying the discount factors and highlight the importance of choosing proper planning horizons with appropriate discount factors. Recent works confirm this by showing that it is beneficial to have an adaptive discount factor [4] or multiple discount factors [21].

Third, failure analysis could identify tasks and states where RL agent's attention drastically differs from expert humans. In a human-in-the-loop RL paradigm, these states may need human intervention or correction. By using our method for comparing human and RL attention, researchers may diagnose their algorithms with the states of their interest.

For researchers who are interested in using RL as models for cognition, it is perhaps both surprising and encouraging to see that RL agents trained from scratch with only images and reward signals can develop attention maps that are similar to humans, especially when considering that they have very little prior knowledge. This result is similar to previous research that shows CNNs trained from image classification tasks can learn features that are similar to the ones in the visual cortex [19, 78, 79]. Consistent with previous findings, we show that model task performance and feature similarity are highly correlated [79]. Our results are complementary to the recent findings using human brain imaging data when playing Atari games [13], suggesting that deep RLs can learn biologically plausible representations and can be used as models for human gaze, decision, and brain activities.

Multiple factors are important for interpreting our results and could explain the remaining differences between human attention and RL attention. The first one regards the nature of human attention. Humans store information in memory and do not need to constantly move their eyes to attend to all task-relevant objects. To complete the human attention map, in addition to gaze data (overt attention), one will need to retrieve human covert attention from brain activity data, a technique that became possible recently [41, 13]. Another factor is the human intrinsic reward. Humans are likely to have internal reward functions that are different from the ones provided by the game environment, and reward is known to affect attention [60, 41]. Hypothetically specific algorithmic or network architecture designs that capture important cognitive aspects of human decision-making could lead to more similar saliency maps.

A closely related research direction compares a human player's *policy* with an RL agent's learned policy [47] which could further allow us to better understand the similarities and differences between humans and RL agents. However, as we have shown here, the difference in decisions could be due to perception, which needs to be considered while comparing policies. Our approach lays the groundwork for future research in this direction.

**Ethics statement** Our work studies how attentional mechanisms of humans and decision-making reinforcement learning agents are similar or different. The hope is that our work will help us better understand the differences and similarities between humans and AI. Comparing humans with AIs allows us to better understand their strengths and limitations. The study involves collecting and modeling human eye-tracking data. We have removed all personally identifiable information such that the private information of human subjects is protected.

## Acknowledgments and Disclosure of Funding

This work is supported by NIH Grant EY05729; A portion of this work has taken place in the Learning Agents Research Group (LARG) at UT Austin. LARG research is supported in part by NSF (CPS-1739964, IIS-1724157, FAIN-2019844), ONR (N00014-18-2243), ARO (W911NF-19-2-0333), DARPA, Lockheed Martin, GM, Bosch, and UT Austin's Good Systems grand challenge. Peter Stone serves as the Executive Director of Sony AI America and receives financial compensation for this work. The terms of this arrangement have been reviewed and approved by the University of Texas at Austin in accordance with its policy on objectivity in research.

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
