# OpenReview forum: "Machine versus Human Attention in Deep Reinforcement Learning Tasks"
_NeurIPS.cc/2021/Conference — NeurIPS 2021 Poster_

### Official Review · Reviewer_bEDj · 2021-07-15

**Rating:** 7
**Confidence:** 4

**Summary:**

The authors investigate the (dis)similarity of the saliency maps of reinforcement learning agents versus visual attention maps of humans performing the tasks of learning to play Atari games for PPO, as well as confirming that the conclusion of positive correlation between the model performance and similarity for other algorithms. The paper poses (and gives some evidence on) the question: do deep RL agents pay attention to the same visual features as humans do?



**Ethical Concerns:**

No concerns.

**Limitations And Societal Impact:**

The paper analyses mostly the performance of PPO (apart from Appendix 6 showing the correlation plots for other methods); the authors acknowledge these limitations and are fair in their assessment of the result. The ethics judgement also seems reasonable to the reviewer, and the data used in the paper appear to be anonymised.

**Main Review:**

*Related work:*

The related work covers different aspects of previous work and acknowledge dual aspect of this work, namely explainable reinforcement learning and studies of human attention.

*Originality and significance*

While the work compares (mostly) the existing PPO method using existing saliency maps for interpretable reinforcement learning (RL), it gives a valuable insight about the relation between human and RL attention for PPO methods.  The work lies at the intersection of interpretable RL and human attention, and it would help build upon current understanding of failure modes of the RL methods such as PPO. It could be also extended to the other methods to see whether the same results are maintained for different RL algorithms.

*Pros:*

 The paper studies the perceptional difference between estimated human attention and RL agents  and links it to the failures of control of the RL agents. This is an important problem, and the authors shed the light on it using a well-defined methodology
The paper is clearly written (although some outstanding questions remain) and gives a comprehensive overview of the state-of-the-art methods. The paper demonstrates the relation between human and PPO model attention in a case of different hyperparameters and  tasks; it also sheds a light on what happens during the failure state.

*Cons:*
The experiments are carried out mostly for the PPO algorithm (except for correlation analysis in Appendix 6 which only shows that the policy improvement leads to better correlation of human vs RL attention); it would be interesting to know more about how it generalises to other methods but I accept the authors explicitly propose it to be a part of the wider study and the current work has enough merit, in a view of the reviewer

*Other comments:*
[1] "In order to capture variance in training and ensure reproducibility, we select six popular 119 Atari games instead of using all Atari games and train 630 agents in total (300 GPU days on GeForce 120 GTX 1080/1080 TI).” How many agents per each game, is it 105? Are settings the same?

[2] “The unseen states are late-game states obtained from human experts’ data [81] which RL agents have not encountered.” The reviewer thinks would be good to clarify what the word late would mean in this context, i.e. how exactly these late-game states have been selected, was it some sort of threshold on the frame number or something else?

[3]One of the potential limitations for the most of the experiments is that they compare RL and *predicted* human attention. It has been addressed in the appendix 7 by extracting human fixations and it shows that the fact that the conclusions stay the same (from Appendix 7: *In general, the agents’ attention becomes more similar to humans’ as training progresses* ) for the actual gaze  collected using the EyeLink post-processing software to extract the fixations. The reviewer is wondering on whether it was collected on the whole data[81] or what was the subset of data? Also, it was the reviewer's impression from the main text that the dataset contained gaze fixations and you performed the attention map prediction using convolutional-deconvolutional network based on these fixations, so why was EyeLink software needed for the experiment in Appendix 7 but not for the main text?

[4] "Fig. 4c shows the (negative) KL histogram of 100 failure states vs. normal states in which the human and RL agents are more likely to agree on the objects to be attended to.” It sounds like an interesting case. In the appendix, it is shown that while the attention map makes sense, the failures are caused by the policy. Would it be possible to clarify whether it was caused by the stochasticity of the policies or do the authors see any other explanation?

[5]  Fig. 1: would it be possible to reference to the exact methods used for control attention models (optical flow and saliency) as the reviewer would struggle to reproduce this experiment just from the text?

**Update**

Score increased to 7 as a result of rebuttal discussions as the authors cleared a number of concerns (see below).


**Time Spent Reviewing:**

6

---

> ### Author Response · Authors · 2021-08-10
> **Thank you!**
>
> Thank you for your encouraging feedback! Regarding your questions and comments:
>
> > [1] "In order to capture variance in training and ensure reproducibility, we select six popular 119 Atari games instead of using all Atari games and train 630 agents in total (300 GPU days on GeForce 120 GTX 1080/1080 TI).” How many agents per each game, is it 105? Are settings the same?
>
> Yes, we trained 105 agents per game. And the settings are the same except: 1) the random seed (for consistency check), and 2) the hyper-parameters we vary.
>
> > [2] “The unseen states are late-game states obtained from human experts’ data [81] which RL agents have not encountered.” The reviewer thinks would be good to clarify what the word late would mean in this context, i.e. how exactly these late-game states have been selected, was it some sort of threshold on the frame number or something else?
>
> Thanks for the question. We use a score threshold, i.e., we select human frames with scores that are much higher than RL best scores (on average 50x better). We will add the exact threshold in our published data and in the appendix.
>
> > [3]One of the potential limitations for the most of the experiments is that they compare RL and predicted human attention. It has been addressed in the appendix 7 by extracting human fixations and it shows that the fact that the conclusions stay the same (from Appendix 7: In general, the agents’ attention becomes more similar to humans’ as training progresses ) for the actual gaze collected using the EyeLink post-processing software to extract the fixations. The reviewer is wondering on whether it was collected on the whole data[81] or what was the subset of data? Also, it was the reviewer's impression from the main text that the dataset contained gaze fixations and you performed the attention map prediction using convolutional-deconvolutional network based on these fixations, so why was EyeLink software needed for the experiment in Appendix 7 but not for the main text?
>
> A very good point. The fixations for the experiments in Appendix 7 are extracted for all human data. The fixation extraction needs to be done when using human data, because a few frames in human data do not contain meaningful gaze data (e.g., due to blinking). Fixation extraction excluded these frames. For the main text, we were training a gaze predictor which can use the raw gaze data as in [81].
>
> > [4] "Fig. 4c shows the (negative) KL histogram of 100 failure states vs. normal states in which the human and RL agents are more likely to agree on the objects to be attended to.” It sounds like an interesting case. In the appendix, it is shown that while the attention map makes sense, the failures are caused by the policy. Would it be possible to clarify whether it was caused by the stochasticity of the policies or do the authors see any other explanation?
>
> This could be due to the stochasticity of the policies.
>
> > [5] Fig. 1: would it be possible to reference to the exact methods used for control attention models (optical flow and saliency) as the reviewer would struggle to reproduce this experiment just from the text?
>
> Thanks for pointing this out. We will add references (to paper and code) for these baselines.

---

> > ### Comment · Reviewer_bEDj · 2021-08-27
> > **Thank you for the responses**
> >
> > Thank you for the responses!
> >
> > [1] That sounds good, it'd be useful to have it in the revised version when possible.
> >
> > [2] Agreed, would be good to add the description
> >
> > [3]  I think it makes sense to clarify these points in the paper (potentially appendix), especially on justification of fixation extraction. Relate to that, after reading the comments about the gaze prediction from other reviewers, I am wondering if there should be some improved justification of semi-frame-by-frame model in the revised paper. I see that it remains the main sticking point. From one hand, I understand from the paper that it improves the quality, from another hand, there is a high risk that the human attention becomes different as the humans are solving different task. How would the authors justify that the conclusions and experimental claim were not affected by this difference? It seems to me the principal remaining sticking point for me after reading the reviews.

---

> > > ### Comment · Reviewer_bEDj · 2021-08-27
> > > **Continued response**
> > >
> > > [4] {Not critical but might be one possible presentation suggestion to confirm whether due to the stochasticity the policies can attend to wrong objects} I could imagine it could be quite good to show (maybe in the appendix?) hat multiple different runs with the same input but different seeds showing that it's the stochasticity of the policy which causes it.
> > > [5] Adding it to the revised version sounds good.

---

> > > > ### Author Response · Authors · 2021-08-29
> > > > **Follow-up Response**
> > > >
> > > > Thank you for the follow-up comments. We will include the clarifications and figures in the revision.
> > > >
> > > > On [3], a simpler way to think about this is that the RL agents are essentially playing the games in the same semi-frame-by-frame fashion, since they take an action every 16 frames (as determined by the preprocessing protocol) of the game. In this sense, the original game speed for human players is the real “different task”. Except for the slower speed, all other aspects of the games (e.g., action manipulation and reward structure) are the same in our dataset. The human subjects have the same behavioral goal, with less rushed--and thus more precise--attention allocation, decision making, and action execution. This will lead to some difference in their attention distribution, but not because humans are solving a different task, but because they are allowed enough time to really process the visual input. We hope this resolves your question :)

---

> > > > > ### Comment · Reviewer_bEDj · 2021-08-31
> > > > > **Increasing to 7**
> > > > >
> > > > > After an internal discussion with other reviewers as well as seeing better rationale behind semi-frame-by-frame learning provided by the authors,  and also taking into account the merits of produced analysis, I decided to increase my score to 7.

---

### Official Review · Reviewer_HwvK · 2021-07-16

**Rating:** 7
**Confidence:** 3

**Summary:**

The authors present a comparison of a model trained on human gaze data when playing Atari games and a PPO agent in terms of saliency maps. The authors run a large number of experiments and identify a number of interesting results in terms of where the saliency maps match and do not match.

**Limitations And Societal Impact:**

All major limitations of the paper are addressed.

**Main Review:**

Originality: As the authors state, they are the first to undertake research like this, to the best of my knowledge. The results are also novel, though many reaffirm findings from elsewhere. Overall, the work is of sufficient originality for inclusion.

Quality: The work is of very high quality. The authors present deep, in-depth experiments and rigorous analysis. The appendices are particularly helpful here and would be especially pertinent for future researchers. My only concern when it comes to quality is the relative lack of information on the human gaze model. While this is understandable given space limitations, there's a few details I'd like more detail on. First, how was data split? The authors state they used an 80-20 split, but with no further details. If it was by a random sampling I would have concerns, as it's possible the same human individuals ended up in both the training and test datasets, which could skew the metrics in favour of the model. Second, the authors are quite vague on the model and the training process for it, stating only "we follow their approach to train models with human gaze data". Finally, I'd appreciate more detail on the dataset in question.

Clarity: I have a couple of concerns when it comes to the clarity of the work. First, the authors do not make clear until page 3 that they aren't directly comparing RL saliency maps and human gaze data, While in hindsight I can identify why this would be challenging, the authors never make the case why this is necessary. Throughout the paper the authors refer to their work as comparing humans and RL agents, instead of a proxy for humans. There are also some oddly worded sentences that I can't quite parse. For example "By publishing our human gaze prediction model researchers can diagnose their algorithms with the states of their interest.".

Significance: I think any RL researcher would be interested in this work, with particular interest coming for researchers who employ the Atari domain. The findings are novel and valuable with several promising directions for future work.

**Time Spent Reviewing:**

1

---

> ### Author Response · Authors · 2021-08-10
> **Thank you!**
>
> Thank you for your encouraging feedback! Regarding your questions and comments:
>
> > My only concern when it comes to quality is the relative lack of information on the human gaze model. While this is understandable given space limitations, there's a few details I'd like more detail on. First, how was data split? The authors state they used an 80-20 split, but with no further details. If it was by a random sampling I would have concerns, as it's possible the same human individuals ended up in both the training and test datasets, which could skew the metrics in favour of the model. Second, the authors are quite vague on the model and the training process for it, stating only "we follow their approach to train models with human gaze data". Finally, I'd appreciate more detail on the dataset in question.
>
> For each game we have 20 trials of data (15 min long per trial). We randomly select 16 trials for training and 4 for testing. This makes sure that the data belonging to the same trajectory will not end up in both training and testing. We will add more details for the human gaze model and dataset in Appendix 1, in addition to what we have.
>
> > I have a couple of concerns when it comes to the clarity of the work. First, the authors do not make clear until page 3 that they aren't directly comparing RL saliency maps and human gaze data, While in hindsight I can identify why this would be challenging, the authors never make the case why this is necessary. Throughout the paper the authors refer to their work as comparing humans and RL agents, instead of a proxy for humans.
>
> You raised a very important point here. We tried to explain this issue called “state distribution mismatch” in line 96-101. The human data and RL data are different due to the difference in their policies. Having a gaze prediction model (trained on human gaze data) allows us to compare attention in new states generated by the RL agent, so we are no longer limited to human states in [81]. We also conducted experiments using the ground truth human gaze (without prediction) and the results show a similar trend as presented in the main results (line 334 and whole Appendix 7). We will emphasize this early on in the revision.
>
> > There are also some oddly worded sentences that I can't quite parse. For example "By publishing our human gaze prediction model researchers can diagnose their algorithms with the states of their interest.".
> We meant to say that “By using our method for comparing human and RL attention, researchers can diagnose...”. Thank you for pointing out the typo.

---

### Official Review · Reviewer_hVFS · 2021-07-16

**Rating:** 7
**Confidence:** 4

**Summary:**

This work proposes to use human gaze data and an attention model derived from it to compare and contrast human attention and attention saliency maps extracted from PPO deep RL agents on Atari Learning Environment games. The authors analyze the effect of learning, discovering that learning improves similarity between agent attention and human attention. The authors evaluate how the discount factor contributes to similarity, finding that attentional similarity and performance peak simultaneously, with fairly high discount factors. The authors also perform some focused analyses of how attention varies in failure states for these artificial agents, and how attention behaves when attempting to generalize to unseen data from the same games.

**Limitations And Societal Impact:**

I agree with the authors’ assessment of the minimal direct social impact of their work and find their position on comparing humans and artificial agents compelling.

However, the language of the abstract and the discussion frames their findings as regarding deep RL agents at large, while the evidence provided is limited to a single architecture. The results provided in Section 4.5 do not, to the best of my understanding, offer evidence that the conclusions from the analyses would generalize to other architectures; only that it would be feasible to perform the same analyses on other model architectures. I find this to be perhaps the work’s most significant limitation, and it is not discussed in the paper.

**Main Review:**

**Post-discussion notes:** Following the authors' responses and discussion with the other reviewers, I'm revising my rating to 7. I still believe there's room to ablate some of the decisions on how you constructed the metric, but I believe it captures something meaningful as-is, alleviating my primary concern with the paper.

**Originality:** I find the approach interesting and original, clearly building on the gaze-related work of [81], and expanding to perform meaningful success and failure analyses of PPO deep RL agents. This work makes intriguing use of the human data and model derived from it.

**Quality:** While the work seems to have been approached with rigor in mind, I have several questions regarding some decisions made, that impact my confidence in the rigor of the paper as a whole. One issue is the choice of loss on the saliency maps — given that the entire set of qualitative analyses relies on the saliency metric, additional clarity on different choices made with the saliency metric would be quite helpful [see additional details below]. In addition, several of the claims made in sections 4.3-4.5 appear to be generalizations from a single observation (or if they are from visually examining a larger set of examples, it is not made clear). Providing metrics to quantify some of these effects (4.2, 4.4) would go a long way in providing rigor. Finally, while I understand the technical limitations underlying the decisions made in section 4.5, providing evidence that one could replicate this effort to additional model architectures is different from providing evidence that the results of these analyses are likely to generalize to other models.

**Clarity:** While the overall structure of the paper is sensible, I found the paper unclear in several locations. See more detailed notes and questions below. The term ‘feature’ is used for distinct things in Sections 1 and 2, making it harder to navigate. Some detail on the models of human attention (exact inputs and outputs) is lacking. The organization of some of the results sections is odd (see comments below), and it is hard to appreciate if some conclusions drawn were the result of a single visualized image, a visual inspection, or what sort of process.

**Significance:** The work provides interesting and novel data on how attention saliency maps produced by a particular deep RL algorithm align with attention maps produced by a model of human attention on the same data, including analyses of the effect of the course of learning, the discount factor, and several tailored analyses of failure states and generalization. These methods may prove interesting for others developing methods and evaluating them on ALE games, at least for the games for which the authors provide a trained human attention model. Building on this work for other domains or tasks would provide substantially harder, as it would require collecting eye-tracking data for the specific domain (and training an attention model on it).

I look forward to reading the authors' responses and additional reviews.

### Detailed Notes

* **Sections 1/2:**
    * The term ‘feature’ is used often and appears to refer to different aspects of the problem. The abstract and introduction discuss the features learned by humans and machines, that is, some representation or transformation of the visual input. Later in the abstract, the authors refer to ‘visual features’ of Atari games, and in the related work section, discuss the human ability to to ‘move their foveae to the right place at the right time to attend to important features,’ referring to aspects of the input. I would suggest the authors pick different terms to refer to each of these distinct things (as one is a learned representation of the other), as that would clarify some of the language early in the manuscript.
    * Line 77: ‘alternating the input’ — should be altering?
* **Section 3:**
    * Lines 89-95: the authors mention that the human gaze data used was collected (in [81]) in a semi-frame-by-frame mode, which allows human participants to process the input for as long as they’d like before selecting an action. Given the ability to process each frame for as long as they desire, is the comparison between agents and humans in this case (even for the purposes of extracting attention) valid? Doesn’t this imply that the agents and humans are solving somewhat different problems?
    * Lines 102-112: models of human attention — the authors could consider clarifying what the output of these models are — do they produce a saliency map from the gaze data and input frame? If not, what do they produce? Understanding this detail should not require jumping through the references.
    * Line 125: the choice of a loss for the saliency map — is there a particular reason why the L2 loss is chosen? Assuming the policy is normalized to a valid distribution, might a divergence metric (KL, etc.) make more sense? Two follow-up questions about the choice of metric:
        * Not all changes to the policy are equally distinct. Imagine the case where the pre-perturbation policy is 100% move left. If the post-perturbation policy is 100% move right, that makes for a substantially larger change than if the post-perturbation policy is 100% move up and left simultaneously. However, these two changes entail the same L2 distance. Perhaps the metric should take into account the functional effect of the actions, and not merely the change in distribution?
        * Another issue, in some ALE games, comes with the FIRE action, which can be coupled with any other action (see the action space defined in constants here: https://github.com/openai/atari-py/blob/master/atari_py/ale_interface/src/common/Constants.h). While many games declare the full action space, if firing is meaningless in a game, it could be the case that there exist actions that are functionally identical (for example, PLAYER_A_UP and PLAYER_A_UPFIRE) but are represented by different elements in the policy vector, such that shifting mass from one to another retains the same functional policy. Is this an issue with any of the games evaluated?
    * Lines 139-147: given the fact the humans can attend to as much of the image as they would like to, while the agent is limited to a single pass, was the opposite KL divergence (KL QP) considered, that would only penalize the agent if it attended to regions the humans did not attend to?
    * This might require combining all fixations/gaze samples for each player and input image to a single input, which does not sound unreasonable, but I am not aware whether or not [81] performed such preprocessing.
* **Section 4:**
    * Lines 155-156: perhaps I missed this detail in other sections of the paper, but — given the standard approach of stacking four consecutive frames, but the human having access to a single frame at a time to view for as long as they desire (and hence, their gaze is presumably less influenced by previous frames), why is it sensible to compute the human saliency maps from a set of four consecutive images?
    * Lines 162-169: the consistency of the saliency maps across random seeds seems a little surprising. Does this occur often with saliency maps in computer vision or similar approaches in language processing, or do those show greater variability across random seeds?
* **Section 4.1/Figure 1:**
    * The text should be much larger. If the manuscript were to be printed out, it would be near-impossible to read. This comment stands for all figures.
    * Is the comparison to motion and saliency (the green and red lines) between the RL agents and these alternative methods or the human data and these alternative methods? I assume it is the former, but perhaps the caption should clarify.
    * Is there a hypothesis for why the human comparisons show a minimal gap between the CC and the KL, but the other two methods show a substantial gap between the two evaluation metrics?
    * Lines 186-193: I am not sure I understand the statement regarding the correlation values between CC/KL and game scores for the baselines. Examining the data in appendix 2, figure 1, for Breakout, the correlations between game score and motion, and game score and saliency, are statistically significant when using cross-correlation. Could this statement be clarified?
* In Figures 3-5, when human attention is plotted, is it the output of the human gaze model? Or an example of a gaze from the human data? If it is plotting the output of a model of human attention, I would suggest changing the caption to ‘Human Model’, rather than ‘Human’, which implies human data is plotted.
* **Section 4.2/Figure 3:**
    * The figure plots a single example from two games, but the claims made appear to be broader. Beyond these examples (and examples in the appendix), could there be a metric for the number of objects the agents attend to? For example, the number of clusters/contiguous regions in the saliency maps produces by the agents? Computing a metric from more examples would help substantiate the point on the number of attended objects varying strongly as a function of $\gamma$.
    * The caption to Figure 3(b) mentions that “the human pays attention to an approaching enemy from the left and oxygen level at the bottom,” but the human panel only shows attention to the enemy on the left (and mildly to the agent itself). What supports the claim that the human pays attention to the oxygen bar? If additional gaze locations during this frame, it might be worth superimposing them somehow? As it is, the caption does not match the visual.
    * Lines 215-217: the claim that attending to objects that matter in the long term is likely to be captured by covert attention, rather than overt attention — what is it based on? Is there any evidence for this claim, including negative evidence (e.g. not attending to the igloo in Frostbite for most of the gameplay, even though it is necessary to finish a level)?
* Section 4.3/Figure 4:
    * Minor narrative question: I found it mildly confusing that the discussion begins from the latter parts of the Figure — section 4.3 starts from discussing subfigures (d) and (e), and only then move to (a-c). It might be a little clearer to organize the subplots and the discussion in the same order.
    * 4(d): is the implication of this figure that sometimes when the agents fail they attend to the same objects, and in other times when the agents fail, they attend to different objects? If I follow, it demonstrates that attention mismatch is neither necessary nor sufficient for failure.
    * The caption mentions both significances at the 0.05 level (*) and at the 0.01 level (**), but all results reported in this figure are significant at the ** level. It could be clearer to omit the description of significance at the 0.05 level, then? Or am I missing the use of the single * somewhere? [this appears to only be used in Figure 5, in which case I might suggest mentioning it there].
    * Assuming I understand them, I find the empirical results reported in this section compelling — the degree to which agent attention and human attention match at failure states varies by games, in some games matching better in failure states, while in other games matching worse in failure states. However, I’m lacking any conclusion from this section, or any testable predictions or hypotheses. What is common between the games in which failure state attention is more similar, or less similar, or the games in which there appears to be no change in the distribution? Does this tell us something about what might be optimal in different games, or when matching human attention might be a better or worse predictor of success?
* **Section 4.4/Figure 5:**
    * Similar to the question for Figure 3 — a single example is plotted and followed with a more sweeping claim (similarities in Frostbite and Montezuma’s Revenge drop due to new objects the agents have never encountered). Is there evidence this statement generalizes beyond the plotted example? Was it made by inspecting all failure states? If it is important, I imagine it could be supported in a more quantitative way? e.g. locate the sprites of the new objects (matching single RGB pixels can be sufficient with Atari games, as they tend to use different colors), and compute the similarity between a saliency map computed from new objects only and the attention in these failure states.
    * Lines 277-281: while I agree with the spirit of the conclusion, framing these results as locating ‘the obstacle’ between these agents and expert human-level performance seems rather ambitious. Perhaps ‘an obstacle? e.g. the other components [37] points to could prove equally important (if not more so).
* **Section 4.5:**
    * If Montezuma’s Revenge is excluded from further analysis because the other RL agents fail to learn it, why is it a good candidate for analysis using PPO?
    * I find this section a little sparse — there is little doubt that the analyses performed for PPO _can_ be done for these other algorithms — but the lack of performing even a single one (for example, the analyses in figure 1, of the impact of learning on the similarity, or of the discount factor) does not provide any evidence that the result from analyzing other algorithms will be similar to the results with PPO.
    * I understand the technical difficulty — performing these analyses would require either training the models, which is computationally expensive,  or receiving checkpoints at different intervals in training, which may not exist. However, as it stands, Section 4.5 offers little evidence that the results of the (interesting and meaningful) analyses on PPO will transfer to other algorithms, especially state-action value algorithms, due to the difference mentioned in the Appendix.
* **Section 5:**
    * Line 298: ”We have also identified further challenges in closing the performance gap between human experts and RL agents.” — I was a little surprised the text does not reiterate what these challenges are.
    * Lines 311-314: “Our results indicate that RL agents first learn to attend to a few important objects like humans, then learn to attend to other objects.” This does not seem abundantly obvious from the results plotted in Figure 2, which as far as I can tell is the only attempt to visualize how many objects the agents attend to at different points in training? Is there any quantitative support for this point? See my question/suggestion regarding Figure 3 as well.

**Time Spent Reviewing:**

7-8

---

> ### Author Response · Authors · 2021-08-10
> **Thank you!**
>
> Thank you for your very detailed review. We truly appreciate your thoughts and suggestions. Regarding your questions and comments:
> - The feedback on the figures, captions, and narrations will be updated as suggested.
> - On the general comments on Figure 3, Figure 5, and Section 5, about the example frames:
> 	- Our conclusions are all based on the statistics we show on the aggregated data (e.g., Figure 1 and 4a). The plotted frames are representative examples of these results. Thank you for the suggestion of analysis on “the number of clusters/contiguous regions in the saliency maps produced by the agents”. We will perform this in the revision.
>
> To maintain the structure and visibility, we mark our reply for the rest of the questions in **bold**:
> - Sections 1/2:
> 	- The term ‘feature’ is used often and appears to refer to different aspects of the problem...
> 		- **Thank you for the pointing this out. On a more conceptual level we are interested in the "representations" learned by human and RL agent. Here in the paper, we are comparing saliency maps, and thus "features" refer to aspects of the input as in objects and patterns.**
> - Section 3:
> 	- Lines 89-95: [Regarding the human dataset] ...is the comparison between agents and humans in this case (even for the purposes of extracting attention) valid?
> 		- **Given enough time to process the input, human players were able to provide more accurate eye movement data that correctly indicate behavioral relevant objects (for extracting attention). In some ways, this is more similar to the task for the agents since RL agents take an action for every frame (as selected during processing).**
> 	- Lines 102-112: models of human attention... do they produce a saliency map from the gaze data and input frame?
> 		- **Given an input frame, the human attention model produces a heatmap of the same dimension that predicts human gaze distribution. This model is trained using human gaze data as the ground truth. We will clarify this in the revision.**
> 	- Line 125: the choice of a loss for the saliency map — is there a particular reason why the L2 loss is chosen?
> 		- **L2 is chosen because this is the general loss that can be used for policy as well as value (e.g., for DQN).**
> 	- Not all changes to the policy are equally distinct. Imagine the case where the pre-perturbation policy is 100% move left. If the post-perturbation policy is 100% move right, that makes for a substantially larger change than if the post-perturbation policy is 100% move up and left simultaneously. However, these two changes entail the same L2 distance. Perhaps the metric should take into account the functional effect of the actions, and not merely the change in distribution?
> 		- **The functional effect of the actions is determined by the environment transition and reward functions. In your example, if move right / move up and left both result in the same expected return, their loss should be the same.**
> 		- ...if firing is meaningless in a game, it could be the case that there exist actions that are functionally identical... but are represented by different elements in the policy vector, such that shifting mass from one to another retains the same functional policy. Is this an issue with any of the games evaluated?
> 			- **We have confirmed that all training environments that include the "FIRE" actions do have firing as a meaningful action. Thank you for the observation.**
> 	- Lines 139-147: given the fact the humans can attend to as much of the image as they would like to, while the agent is limited to a single pass, was the opposite KL divergence (KL QP) considered, that would only penalize the agent if it attended to regions the humans did not attend to?
> 		- **The targets of human gaze are usually only a subset of all behaviorally relevant targets on the screen. In most cases the RL agent attends to many more locations than humans do. Therefore it is more important to penalize the agent for not attending to human-attended locations.**
> 	- ...combining all fixations/gaze samples for each player and input image to a single input...
> 		- **A good point--however the human data was collected during realtime game play, so the frames are not necessarily the same each trail, even for the same player. In this case, the human attention model shows another advantage--trained on a majority of human gaze data (training set), it predicts the distribution of gaze rather than a single location for an input frame.**
> - Section 4:
> 	- Lines 155-156: ...why is it sensible to compute the human saliency maps from a set of four consecutive images?
> 		- **Human players do not make a majority of decisions on a single frame. In fact, human attention and visuo-motor decisions are heavily based on motion information (e.g., direction and speed of motion) embedded in consecutive frames. Therefore human saliency maps must be produced from consecutive images that indicate motion.**
> 	- Lines 162-169: the consistency of the saliency maps across random seeds seems a little surprising. Does this occur often with saliency maps in computer vision or similar approaches in language processing, or do those show greater variability across random seeds?
> 		- **To the best of our knowledge vision and language studies do not check consistency between repeatedly trained models with different random seed (e.g., [13,36]). Because in these tasks training data are fixed so there is less stochasticity in the trained models as long as they converge. In our case, the training process of RL is inherently stochastic so this consistency check is necessary.**
> - Section 4.1/Figure 1:
> 	- Is the comparison to motion and saliency... between the RL agents and these alternative methods...?
> 		- **Yes, all comparisons are made with RL agent attention. We will clarify in the revision.**
> 	- ... why the human comparisons show a minimal gap between the CC and the KL, but the other two methods show a substantial gap between the two evaluation metrics?
> 		- **CC and KL have very different semantics and are not directly comparable. This is just a coincidence.**
> 	- Lines 186-193: ...the statement regarding the correlation values between CC/KL and game scores for the baselines. Examining the data in appendix 2, figure 1, for Breakout, the correlations between game score and motion, and game score and saliency, are statistically significant when using cross-correlation...
> 		- **This sentence was referring to the aggregate results for all games. You are right about game Breakout, but for other games in general motion / saliency do not correlate well with game scores.**
> - Section 4.2/Figure 3:
> 	- Lines 215-217: the claim that attending to objects that matter in the long term is likely to be captured by covert attention, rather than overt attention — what is it based on?
> 		- **Due to the limitations of human attention, objects that matter in the long term are often retained in the working memory, and are revisited by gaze at lower frequency (Theeuwes et al., 2009; Johnson et al., 2014).**
> - Section 4.3/Figure 4:
> 	- 4(d): is the implication of this figure that sometimes when the agents fail they attend to the same objects, and in other times when the agents fail, they attend to different objects?
> 		- **The idea here is precisely that failure could be caused by problems in two different stages: in some cases, the agent does not learn to attend to the correct objects; in other cases, it does not learn the optimal policy even with good perception. This distinction could be used to diagnose failing agents in other scenarios.**
> 	- I’m lacking any conclusion from this section, or any testable predictions or hypotheses...
> 		- **Using our method we are able to qualitatively identify the above two types of failure states. However, with such small number of games tested, it is difficult to draw conclusions on the characteristics of the games based on this statistics. Follow-up studies are looking at all games within the human dataset, which should provide a better insight for your question.**
> - Section 4.5:
> 	- If Montezuma’s Revenge is excluded from further analysis because the other RL agents fail to learn it, why is it a good candidate for analysis using PPO?
> 		- **Here we are calculating the correlation between algorithm performance and attention similarity. So if these RL agents all have zero scores the calculation cannot be done. But zero scores do not prevent us from performing failure analysis, for example.**
> 	- I find this section a little sparse... Section 4.5 offers little evidence that the results of the (interesting and meaningful) analyses on PPO will transfer to other algorithms...
> 		- **Here we only demonstrate that the positive correlation between game scores and attention similarity can be replicated in other RL architectures. The other detailed analyses (training and discount factors) could be done for those architectures but may not necessarily yield the same results as PPO.**
> - Limitations:
> 	- ...the language of the abstract and the discussion frames their findings as regarding deep RL agents at large, while the evidence provided is limited to a single architecture. The results provided in Section 4.5 do not, to the best of my understanding, offer evidence that the conclusions from the analyses would generalize to other architectures; only that it would be feasible to perform the same analyses on other model architectures. I find this to be perhaps the work’s most significant limitation, and it is not discussed in the paper.
> 		- **Our results in Section 4.5 show that the correlation between game score and attention similarity does generalize to other architectures. We will weaken our claim and emphasize that this correlation does not necessarily imply that the detailed analyses (training and discount factor) will yield a similar trend. However, we want to put forth our analyses as an approach to interpret agents with different architectures in future studies.**

---

> > ### Comment · Reviewer_hVFS · 2021-08-25
> > **Follow-up to author response**
> >
> > Thank you for your detailed reply, and my apologies for the delayed response. I have one larger uncertainty, and a few smaller points that your previous reply didn't clarify if the camera-ready revision would include.
> >
> > My larger concern is regarding Section 3, and specifically the commmet on changes to the policy: I'm not sure I understand how your point regarding the functional value of actions answers my confusion. I agree that if two actions, in the long run, offer similar returns, they should be equally preferred. However, deep RL methods are nototriously sensitive to pixel-level purturbations (see [Qu et al., 2019](https://arxiv.org/abs/1911.03849) for one example). Given the evidence that a single, unrelated pixel can cause a drastic change in policy, I find the change in the policy alone to be a questionable metric. You could examine the impact of blurring the image on the value estimate of the model ("is the pixel meaningful to the valuation"), or examine the expected value of the original and purturbed policies under the original (unpurturbed) value ("does the purturbation make me take a worse action"). *If you have any further thoughts on why the change in policy is a meaningful metric, I would be interested to hear them.*
> >
> > Below are the smaller points I was unsure if you would try to fit into a camera-ready revision if the paper is accepted, and I would encourage to consider (no need to reply on these further):
> >
> > * Sections 1/2, re, the term 'feature': I think you would greatly clarify your introduction if you disambiguated these terms in the camera-ready version, and used each word to refer to a specific concept within this problem (it was unclear if you intended to).
> >
> > * Section 4.1/statement across all games and Breakout results: also worth clarifying in the camera-ready version (was unclear if you intended to).
> >
> > * Section 4.2/working memory and attention: same point re: camera-ready.
> >
> > * Limitations: that change would make sense. Offering the approach as potentially helpful in other cases is very different (and much more within scope) than claiming the findings here would generalize.

---

> > > ### Author Response · Authors · 2021-08-29
> > > **Follow-up Response**
> > >
> > > Thank you for the reminders on the smaller points. We will make the clarifications in the revision.
> > >
> > > For your question regarding Section 3: Yes, we agree that one can examine the impact of blurring the image on the value estimate, which was also done in Greydanus et al. Whether to use policy or value really depends on the algorithm. In our setting, we used a standard PPO, which does not crop out the game score area. We indeed visualized the attention map using the value output, and not surprisingly, the attention is on the game scores which is directly related to value estimation. For this reason, using attention obtained from the policy is more meaningful. As for your question on whether policy alone suffices for the attention metric, we list three reasons below:
> > >
> > > 1. First, the Greydanus method applies a local blurring effect, so the change is smooth and the policy network is less likely to have dramatic changes than if we applied a pixel-level adversarial attack, as in your reference. Hence the method is in general reliable, and particularly when we use the Atari benchmarks, which are the original benchmarks Greydanus et al. considered.
> > >
> > > 2. As part of the original method, our metric is the L2 distance between the latent representations of the policy before and after the perturbations. Although the actions are discrete, there could be drastic differences between the latent representations, and those representations are not likely to be equally distanced. Thus, the L2 distances between the original policy (e.g., going left) and two potential perturbed policies (e.g., up & left vs. right, as in your example) are not necessarily the same, depending on the actions’ functional effect. On the other hand, if two perturbed policies (e.g., up & left vs. right) have similar distances from the original policy (e.g., left), this implies that they are behaviorally similar.
> > >
> > > 3. For policy gradient methods, the gradient is \nabla log pi(a|s) Q(s, a), which means if two actions lead to the same value, then from the policy gradient’s perspective, there's no difference between these two actions.
> > >
> > > To sum up, using the change in the latent policy representation as our metric does take into account the functional effect of the actions, and thus is a meaningful metric even without using the value estimation. We hope this resolves your concern :)

---

> > > > ### Comment · Reviewer_hVFS · 2021-08-30
> > > > **Final discussion response and revised rating.**
> > > >
> > > > Dear authors,
> > > >
> > > > Thank you for the thorough response. I'm not sure I understand why you refer to the policy outputs as latent policy representations, as that _is_ the policy for that state, the distribution of action probabilities. However, that's not particularly important.
> > > >
> > > > Following your responses and discussion with the other authors, I'm revising my rating to 7, a good paper, accept (which I also communicated to the AC). I still believe there's room to ablate some of the decisions on how you constructed the metric, but I believe it captures something meaningful as-is, alleviating my primary concern with the paper.
> > > >
> > > > Thank you for the engaged discussion.

---

### Official Review · Reviewer_sG7Q · 2021-07-20

**Rating:** 5
**Confidence:** 4

**Summary:**

The paper compares RL agent attention with human attention on various Atari games. Correlations between RL agent and human attention with learning performance, failure states, discounting, and generalization are examined.

**Limitations And Societal Impact:**

Yes, the paper acknowledges that many of its claims are speculative and provides an "Ethics statement" explaining their handling of sensitive data.

**Main Review:**

Originality: Coincidentally two years ago, I wanted to explore the same research question as the paper: "How similar are the visual features learned by RL agents and humans when performing the same task?" In fact the primary domain I was interested in experimenting on was the Arcade Learning Environment, presumably the same set of Atari-based environments used by the paper. However a key obstacle was the lack of relevant human attention data on Atari environments. The paper attempts to address this by using what they denote as "human expert gaze data" from the Atari-HEAD dataset. This falls short for reasons explained in the section labeled "Quality" directly below.

The paper claims that they "present the first study that compares the RL agent's attention with human attention." The paper does present some interesting results on correlations and KL divergences between changes in human and RL attention over training. However the results on failure states and unseen data don't have clear trends, and the papers provides only speculations as to why many of the results are what they are.

Quality: The paper is fundamentally based on comparisons between RL agent attention and human attention, which is inferred from "human expert gaze data" from the Atari-HEAD dataset. Regarding this dataset, the paper states "The original game runs continually at 60Hz [4], a speed that is challenging even for professional gamers. Human eye movements were rushed and inaccurate at this speed hence could not serve as a useful reference. In [81], however, the human data were collected in a semi-frame-by-frame mode, a design that allowed enough time for the players to attend to all relevant objects on the screen and make decisions. The slowed speed allowed players to achieve world expert level performance [81] which is about 35x better than the human performance reported in previous deep RL literature [43, 26]." Therefore this isn't a comparison involving human attention in a realistic setting but instead a significantly altered setting that could result in markedly different human attention. Though some experiments are interesting, and some show trends in the results, they generally lack robustness: even for the experiments with trending results, it's unclear how the user can benefit from such findings.

Clarity: The paper is well written and easy to understand.

Significance: This has been covered in the sections labeled "Originality" and "Quality".

I believe this paper has good intentions with the work it puts forth, but I believe that the experiments could be made stronger in order to find potentially actionable insights. Furthermore, I believe the paper could dive deeper into its results in order to put forth claims that are supported by resulting evidence instead of putting forth speculations. Maybe most importantly, the paper should consider using human attention data that is representative of realistic settings instead of significantly altered ones unless there are guarantees that human attention remains the same between the two.

=====

[9/1/2021] Update: My response to the Authors' rebuttals can be found below. I'm increasing my rating from a 4 to a 5.

**Time Spent Reviewing:**

4

---

> ### Author Response · Authors · 2021-08-10
> **Thank you!**
>
> Thank you for your constructive feedback! Regarding your questions and comments:
>
>
> > ...the results on failure states and unseen data don't have clear trends...
>
> The results on failure and unseen data don't show consistent trends among all games -- but this is exactly what we want to empathize. From the inconsistencies we can distinguish tasks that are perceptually difficult for RL agents vs. tasks that are not. This helps us understand why the same RL algorithm succeeded on some games and failed on the others.
>
>
> > [Regarding the human gaze dataset] “... this isn't a comparison involving human attention in a realistic setting but instead a significantly altered setting that could result in markedly different human attention.”:
>
> Here we argue that this semi-frame-by-frame setting provides higher-quality human attention data that is more comparable to RL agent attention. As explained in [81], inaccurate human eye movements at the original game speed do not indicate the correct set of decision-relevant objects, which could result in poorer performance in imitation learning and other methods using human reference. Original-speed datasets also encounter the problem of motor delay, where decisions made at one frame are intended for several frames ago. The dataset from [81] eliminates these problems. Based on these reasons, we argue this "significantly altered setting" in fact provides better indication of human attention and thus more reliable results.

---

> > ### Comment · Reviewer_sG7Q · 2021-09-01
> > **Re: Author response**
> >
> > Thank you for addressing the points made in my review! My overwhelming concern regarding experiment design was the use of semi-frame-by-frame human data in comparison with RL agent data, but after reading your response to my review and discussing with the other reviewers, I agree that the use of semi-frame-by-frame data is valid, and likely even more appropriate than using real-time human data, for the purposes of this paper.
> >
> > Regarding trends, I agree that some distinctions in the results from one task to another (e.g., Figure 4) are very interesting and should've emphasized that more in my initial review. Also, I do see that in certain results that don't have emergent trends, the paper does at least try to intelligently speculate how the results came to be.
> >
> > The experiment design and analysis seems solid, and I don't have any strong criticisms regarding these. I think that due to my experience with attention mechanisms, cognitive psychology, and RL, I'm not particularly excited about this work overall and don't see as much promise in this work facilitating future research as many others might. Because you've addressed my concern regarding experiment design, I'll improve my score to a 5. I've shared my reasoning on falling just shy of voting to accept this work with the other reviewers, so they're aware that my issues with this work are more subjective than objective.
> >
> > Thank you for your work.

---

### Decision · Program_Chairs · 2021-09-28

**Decision:**

Accept (Poster)

**Comment:**

UPDATE: The revision has been reviewed and this paper has now been accepted.

----

After extensive discussions among SACs, ACs, and the program chairs, we have decided to conditionally accept this paper.  The crux of the issue is that the paper makes unsubstantiated causal claims about perception based on exploratory analyses (most notably in the final paragraphs of 4.3 and 4.4).  There was some debate around whether this constitutes a "fatal flaw," but it is certainly an issue that needs to be addressed.  The experiments and analyses should be reframed as exploratory first steps.  We hope that the authors will take into account feedback from the reviewers.

Specifically, in order to be accepted, the following conditions must be met:
- The experiments and analyses presented in the paper should be reframed as exploratory rather than causal or confirmatory.
- All unsubstantiated claims and interpretations of results (including those made in the final paragraphs of 4.3 and 4.4, but elsewhere in the paper too) should be removed.

----

The original meta-review for this copy of the paper follows:

This paper compares how humans and machines allocate attention while playing Atari games. The paper utilizes a dataset of human eye tracking measures, and compares a model of human gaze to the attention of RL agents. A number of factors were varied for the RL agents: amount of training, discount rates, whether game states are seen/unseen, etc. This type of work is valuable for interpreting deep RL models and inspiring new algorithms.

Reviewer opinions were initially split. There were a number of issues raised, but the biggest sticking point was the semi-frame-by-frame methodology used in the human Atari-HEAD dataset and whether that was comparable to the operation of the RL algorithms (see R-sG7Q for a summary of the objection). There was substantial back and forth on the issue, both in the public responses and private reviewer discussion.

Although a full consensus was not reached, to summarize the spirit of the discussion, the method used by the authors is at least a plausible one and was justified. In my opinion, the deep RL systems are effectively processing the games frame-by-frame (or close to it), and thus aren't under much "time pressure" in the sense people are at 60 Hz. Thus, data from the human semi-frame-by-frame setting is well-positioned to inform the development of RL models (especially if the eye tracking data is higher quality in this setting).

All things considered, I recommend the paper for acceptance.

In revisions, I hope that the authors will add some of their rebuttal regarding this issue to the paper, to further clarify this decision for other readers. I also agree with @hVFS that the choice of L2 distance in the saliency score (Eq 1) is a little strange, and isn't explained very clearly in the paper or rebuttal.

Overall, the analysis was well-thought out, thorough, and insightful. The open-source tools for comparing RL models with human attention will also be a valuable resource.

**Consistency Experiment:**

NeurIPS has a long history of experimentation. In 2014, NeurIPS ran an experiment in which 10% of submissions were reviewed by two independent committees to quantify the randomness in the review process. This year, we repeated a variant of this experiment to see how the quality of the review process has changed over time.  This paper was part of the experiment and was therefore assigned to two committees (consisting of reviewers, an Area Chair, and a Senior Area Chair) that reached independent decisions.  If both committees made the same recommendation, this recommendation was followed. If a single committee recommended acceptance, the paper was accepted (with the exception of a few cases in which the other committee identified what we considered a fatal flaw, e.g., an error in a key result).

This copy’s committee reached the following decision: **Accept (Poster)**

The other committee assigned to the paper recommended **Reject**.  You can find the other set of reviews, along with any follow up discussion with the authors here:
https://openreview.net/forum?id=8fztRILSxL